# Predicting Tobacco and Alcohol Consumption Based on Physical Activity Level and Demographic Characteristics in Romanian Students

**DOI:** 10.3390/children7070071

**Published:** 2020-07-02

**Authors:** Georgian Badicu, Seyed Hojjat Zamani Sani, Zahra Fathirezaie

**Affiliations:** 1Department of Physical Education and Special Motricity, Faculty of Physical Education and Mountain Sports, Transilvania University of Braşov, 500068 Braşov, Romania; 2Physical Education and Sport Science Faculty, University of Tabriz, 29 Bahman Blvd, Tabriz 51666-16471, Iran; hojjatzamani8@gmail.com (S.H.Z.S.); z.fathirezaie@tabrizu.ac.ir (Z.F.)

**Keywords:** questionnaire, exercise, relationship

## Abstract

Background: This study aims to assess the relationships between alcohol and tobacco consumption, physical activity (PA) and demographic characteristics in Romanian students. Methods: There were 253 participants in this study (112 male and 141 female, age 19.2 ± 0.70 yrs; BMI 22.4 ± 2.2 kg/m^2^), students of Transilvania University of Brasov. The International Physical Activity Questionnaire Short Form (IPAQ-SF), Alcohol Use Disorders Identification Test (AUDIT) and Fagerström Test for Nicotine Dependence (FTND) were employed for the data collection. Results: Results showed that moderate consumption of tobacco and harmful consumption of alcohol had high prevalence among age, gender, year of study and PA level categories. Although the correlation coefficient showed no significant relationship between tobacco and alcohol use, it was shown that a negative correlation between tobacco and PA level existed. Further, age, year of study and PA level had a negative significant relationship with alcohol use among students. In the end, the tobacco and alcohol consumption prediction model showed heterogeneous coefficients. Conclusion: Full models of tobacco and alcohol use were differently predicted by variables, so PAL (Physical Activity Level) could predict tobacco consumption but not alcohol.

## 1. Introduction

It is well known that low levels of physical activity (PA) or the lack of PA can lead to a remarkable increase in risk factors [1]. On the contrary, a higher level of PA plays an important role in improving the quality of life [2].

The World Health Organization recommends 150 min of moderate-intensity aerobic PA throughout the week or 75 min of vigorous-intensity aerobic PA throughout the week or an equivalent combination of moderate- and vigorous-intensity activity, in relation to PA and its benefits on health in adults [3,4]. In this regard, PA has positive effects on health [5,6], as it builds strong muscles and joints, it reduces the risk of developing certain chronic diseases [7,8,9], it raises self-esteem and confidence and it reduces stress and anxiety [4,10]. If we also consider the positive impact that exercise has on academic performance, [11,12,13] and the factors that influence student achievement—cognitive ability and aptitudes (concentration, memory, speaking ability), school behavior (attendance, time allocation for tasks, behavior in society) and academic success (grades, evaluations, exams)—it is quite clear PA is paramount [5]. Additionally, several studies confirmed that academic performance is improved when theoretical classes are reduced in order to increase attendance to classes of physical education [3,8,9,14,15].

Sadly, nowadays, despite all this evidence, many young people ignore recommendations on PA [3], or they do not exercise at all. Instead, they are rather focused on the use of mobile phones [16], tablets or other electronic devices that may be harmful for health.

Raitakari [6] highlighted that adolescents should pay more attention to PA, as individual behaviors of one’s lifestyle are established during this time [6]. In this regard, Seo et al. [13] suggested that smoking and alcohol consumption are two of the most widespread behaviors that compromise health during adulthood [13]. Tobacco consumption, alcohol consumption and the lack of exercise are part of the lifestyle of young adults and represent the main causes of death [14,15]. In Romania, over half of the deaths in adults can be attributed to a series of behavioral risk factors such as: tobacco consumption, alcohol consumption and low levels of PA (62%). This number is well above the EU average (44%) [17]. Further, a better understanding of smoking and drinking behaviors in young adults that may co-occur with PA will help practitioners enhance PA promotion efforts [13]. Several studies have been conducted in relation to and with the purpose to evaluate the predicted factors of drinking or smoking, to better understand the antecedents of involvement in these behaviors. Research findings have shown that tobacco consumption is strongly associated with alcohol consumption in young adults, especially in those who attend college and begin to start smoking [16]. Other studies have also shown that there is a significant correlation between PA, alcohol and tobacco. For example, Vankim et al. [15] demonstrated in a study with 9931 respondents an association between PA and tobacco. In addition, higher levels of moderate and vigorous PA were associated with higher levels of alcohol consumption and lower levels of tobacco consumption [15]. Further, in a random sample of young adult bar customers, aged 18–29 in San Diego, CA (*N*  =  1150), Portland, ME (*N*  =  1019), and Tulsa, OK (*N*  =  1106), Jiang et al. [18] reported a high smoking rate and a strong relationship between smoking and drinking, regardless of the different bar cultures and tobacco consumption regulations from certain cities involved in the study. Smoke-free bar policies were negatively associated with regular smoking. These policies alone may not be enough to influence the association between smoking and drinking, particularly if tobacco marketing continues in the respective places, or in the absence of programs specifically addressing the co-use of tobacco and alcohol [18].

In general, despite the disparities in the prevalence of these risk behaviors, the associations between the behaviors did not differ substantially between two-year and four-year post-secondary populations. However, there is little research on these specific factors that influence the three variables simultaneously in students.

On the basis of the above, the purpose of the study was to show the relationships between alcohol consumption and tobacco, PA and demographic characteristics in students of the Faculty of Physical Education and Mountain Sports in Brasov.

## 2. Materials and Methods

### 2.1. Subjects and Design

This study was carried out on a sample of 258 students (112 male and 146 female). The participants were first-, second- and third-year students, attending regular classes, at Transilvania University of Braşov. Furthermore, the academic specializations involved in this research were: sport and physical education (*n* = 68), sports performance (*n* = 84) and kinetotherapy and special motricity (*n* = 101).

All students that participated in this study had given their written informed consent before the interview. Written informed consent for participation was obtained after the test protocol and after the purpose of the research was explained in detail. The university ethics committee had also given its approval (approval code is: 09102019). Five participants were excluded because of incomplete data. The final sample consisted of 253 students (44.3% men and 55.7% women, 112 male and 141 female participants, age 19.2 ± 0.70 yrs; BMI 22.4 ± 2.2 kg/m^2^).

### 2.2. Physical ActivityLevel

PA was assessed using the official short form version of the International Physical Activity Questionnaire (IPAQ) [19]. The IPAQ questionnaire comprises seven generic items and measures the varying levels of PA intensity, and daily sitting time. This study considers that PA intensity, along with daily sitting time, estimates the total amount of PA in MET-min/week, and time spent sitting. According to the IPAQ, there are three categories of PA levels with respect to health-related recommendations: “low”, “moderate” and “high” [7,20]. The total amount of physical activity per week, measured in MET-minutes per week (MET-min/wk-1), was calculated as follows: duration × frequency per week × MET intensity. Walking, moderate-intensity and vigorous-intensity PA were also taken into account [7,20]. All questions of this questionnaire are related to activities performed in the previous seven days [7,20]. The characteristics of IPAQ are appropriate for assessing the levels of PA in 18- to 65-year-old adults, in various environments. In this study, the reliability of the IPAQ was satisfactory (Cronbach’s α = 0.80) [19].

### 2.3. Alcohol Consumption

The second questionnaire quantified the alcohol consumption and it is called the Alcohol Use Disorders Identification Test (AUDIT, [21,22]), which was translated to Spanish by Rubio [23]. This 10-item scale is measured using a five-point Likert scale, where 0 is “Never” and 4 is “Daily”, for the first eight items. In addition, the last two items are evaluated using a three-point Likert scale which generates a point score of 0, 2 or 4 points. Responses are added up to produce an overall score in relation to alcohol consumption. The total score is categorized into three groups: 8–15 = hazardous drinking, 16–19 = harmful drinking and 20 or above = high-risk drinking. For this questionnaire, the reliability was acceptable (Cronbach’s alpha of α = 0.75) [22].

### 2.4. Tobacco Consumption

the Spanish adaptation by Villareal-González [24] of the Fagerström Test for Nicotine Dependence (FTND, [25]) was employed in order to evaluate tobacco consumption. This instrument evaluates the number of cigarettes, impulse to smoke and nicotine addiction. It includes six questions. The first four are dichotomous (0 = No and 1 = Yes), and the other two follow a four-option Likert-type scale (0 = Never and 3 = Always). The sum of items ranges between 0 and 10, establishing the level of nicotine addiction. Then, we categorized them into the following groups: 1–2 = low dependence (in the present study, no participant assigned to this group), 3–4 = low to moderate dependence, 5–7 = moderate dependence and 8+ = high dependence. This questionnaire presents an excellent Cronbach’s alpha of α = 0.96.

### 2.5. Data Analysis

Descriptive statistics by tables and figures were used to illustrate the preliminary information of alcohol and tobacco consumption in relation to gender, year of study, age and level of PA categories. Further, because the normality of data distribution was not confirmed between gender groups, the Mann–Whitney U test was used to investigate the differences in tobacco and alcohol use between males and females. Pearson correlation coefficients were also utilized to investigate age, year of study and PA level, and tobacco and alcohol consumption rate. Lastly, two ordinal logistic regressions were used to predict students’ tobacco and alcohol consumption based on the other variables. All analyses were conducted by using the statistical program IBM SPSS Statistics 24 (Armonk, NY, USA) at a significance level of *p* < 0.05.

## 3. Results

Table 1 and Figure 1, Figure 2, Figure 3, Figure 4, Figure 5, Figure 6, Figure 7 and Figure 8 show the average, standard deviation (SD) and number of observations of tobacco and alcohol consumption at the different gender, age, year of study and PA categorizations.

As we can see, moderate consumption of tobacco and harmful consumption of alcohol had high prevalence among all categories.

Figure 9 showed no differences between genders for tobacco use (Mann–Whitney *U* = 3537.00, *p* = 0.434). However, there was a significant difference between genders for alcohol use (Mann–Whitney *U* = 3078.50.0, *p* = 0.031).

Further, the correlation coefficient showed no significant relationship between tobacco and alcohol use. It also showed a negative correlation between tobacco and PA level. Factors such as age, year of study and PA level had a negative significant relationship with alcohol use (Table 2).

Subsequent analysis showed that the full models of this study, both in tobacco and alcohol use, are predicted by other variables in a statistically significant manner (Table 3).

Coefficients of the tobacco consumption model showed that only the PA level has the positive effect on tobacco use, so a one-unit increase in PA from 1 to 2 is associated with a 2.35 increase in the odds of tobacco consumption. Further, a one-unit increase in PA from 2 to 3 is associated with a 1.20 increase in the odds of tobacco consumption (Table 4).

On the other hand, the alcohol consumption model showed a different pattern. Years of study and gender were the predicting factors, so a one-unit increase in the years of study from first to second year was associated with a 2.49 increase in the odds of alcohol consumption. Further, being female was associated with a 1.76 decrease in the odds of alcohol consumption (Table 4).

## 4. Discussion

The present study investigated the prediction effect of PA and demographic characteristics on tobacco and alcohol consumption in Romanian Students. A high prevalence of moderate consumption of tobacco and harmful consumption of alcohol was shown among age, gender, year of study and PA level categories. No significant correlation was found between tobacco and alcohol. Further, a negative correlation was observed between tobacco and PA; age, year of study and PA with alcohol use among students. In addition, tobacco and alcohol consumption were predicted by PA and demographic variables, heterogeneously.

Our results were consistent with Rickwood et al. [26], Davoren et al. [27] and Voigt et al. [28]. They mainly reported high levels of harmful drinking among university students. However, Cuban and Mexican health science students classified as at-risk users [29].

Further, our findings were consistent with El Ansari et al. [30], Dawson et al. [31], Von Bothmer et al. [32] and Lorente et al. [33]. They showed that female students had a lower consumption of alcohol, whereas male students engaged in more risky health behaviors than their counterparts (e.g., alcohol use). Furthermore, our findings were inconsistent with Varela-Mato et al. [34], and Pillonet al. [35], who reported that more female students than male students use alcohol and tobacco. Moreover, Mays et al. [36] showed that alcohol use does not differ between men/boys and women/girls.

Our findings concerning the association between PA and alcohol consumption differs from those of other studies that have demonstrated a relationship between them. Our results showed that the PA level had no effect on alcohol consumption, but Lorente et al. reported that participating in sports at a national or international level and training more than six times a week was associated with reduced daily alcohol consumption in French adolescents [33]. It seems PA at a national or international level has a different effect than PA at the university level on alcohol consumption. It seems that future research should focus on the differences of PA at the international, national and local level, etc., and its impact on alcohol consumption.

About tobacco use, other research findings showed that females’ consumption prevalence and real use was lower [30,34]. However, Pillon et al. [35] showed that more female students than male students use alcohol and tobacco. The levels of alcohol consumption among students of both genders are extremely varied from one study to another [28], the variables associated with drinking are many, and, interestingly, drinking alcohol could be associated with risky health behaviors (tobacco use), but also with positive health behaviors (e.g., greater PA) [29]. It seems that male college students perceived themselves less vulnerable to potential health threats [32,37,38], so being male could be associated with drinking alcohol.

Our results are in line with Peretti-Watel et al. [39], who showed that students who entered competitions at the international level were more likely to smoke cigarettes. Further, Henchoz et al. [40] reported that, adjusted for sport and exercise, PA was positively associated with the risk of smoking cigarettes. It seems there were some key notes. The first one is the different classification of levels of PA. Further, the previous research showed that some sports including fighting, martial arts or capoeira were associated with smoking. However, recent findings showed that recreational PA, attending PE classes and participating in sports were independent protective factors for many cigarette use behaviors but not for smokeless tobacco use [41].

Plenty of studies showed the negative relationship between different types of PA and tobacco use as follows: higher initial levels of participation in sports, athletics or exercising related to lower initial prevalence rates of substance use [42]; any smoking was associated with less exercise [43]; smoking has a negative correlation with leisure PA [44]; sport and exercise was negatively associated with at-risk use of cigarettes [43]; sport participation was related to reduced illicit drug use. A systematic review of longitudinal studies showed that in about eighty percent of the studies, sport participation associated with decreased illicit drug use and also 50% of the studies found a negative association between sport participation and marijuana use [45]. Since in our study, different types of tobacco were considered, the results will be justifiable. However, the age periods [46], social and cultural differences of communities [47], the prevalence and prohibition of certain drugs in different countries and also different types of PA [46] should be considered. Previous research indicated that the type of sport has an important effect on the relationship between sport participation and alcohol consumption [46]. However, its structural causes have not yet been determined. The new finding of this study was the investigation of the different levels of PA, alcohol and tobacco use. Although there were no differences between PA level on tobacco and alcohol consumption, increasing the LPA to MPA caused increased tobacco use. Further, in this present study, we found that age had no effect on tobacco and alcohol consumption. This finding was consistent with a previous research that showed no relationships between cannabis use and individuals transitioning into early adulthood [48]. Albeit, we studied ages from 18 to more than 20, however, this finding was inconsistent with the leveling or maturing out hypothesis that cited a decrease in drug use during late adolescence and into early maturity. This result could be due to the nature of the participants in the present study, all of whom were students of physical education and sports sciences.

The second ordinal logistic regression showed that increasing the years of study and being male was associated with an increase in alcohol consumption. Although our findings were consistent with Evans-Polce et al. [49], many studies showed different patterns. In those studies, a relationship between PA and alcohol use was shown. This relationship was observed in both directions and different types of PA [10,29,36,40,41,50,51,52,53].

Although PA level could not predict alcohol consumption, there was a negative relationship between them. In this regard, it was shown that type of sport [54], team or individual sports [55], endurance or strength sports [54], intensity of PA [42,52,55,56], frequency of PA [53], competitive sport and high-contact sports [54] may affect alcohol use. It seems that PA participation was not a protective factor for alcohol use among physical education and sport sciences students.

Despite the novelty of the present study data, some limitations warn against the overgeneralization of the results. First, we could not collect some socio-demographic data that may (e.g., family history of tobacco and alcohol consumption) influence the results. Moreover, the large drop-out of female subjects that did not have a specific cause could interfere with the sample size and gender ratio of the study. Further, subjective or self-reported and objective assessments of variables could lead to different data to some extent. Since physical education and sport sciences students are always required to participate in PA, the amount and type of PA may vary. All participants in the present study were students who appear to be influential in the obtained results. In fact, several influential variables in this study that have not been studied separately (type of PA) will be very important.

## 5. Conclusions

Overall, this research concludes that moderate consumption of tobacco and harmful consumption of alcohol had high prevalence among age, gender, year of study and PA level categories. Further, female students had a lower consumption of alcohol, but no difference was found between genders concerning tobacco use. In the end, second-year students showed more consumption of alcohol. Therefore, the results of this study could be used to design school-based interventions that target alcohol or tobacco use, although more needs to be known about the factors that influence use. The next wave of research should focus on other variables that could lead to more comprehensive models. As such, future research is needed to clarify the factors for tobacco and alcohol use among students who participate in different sports at different levels.

## Figures and Tables

**Figure 1 children-07-00071-f001:**
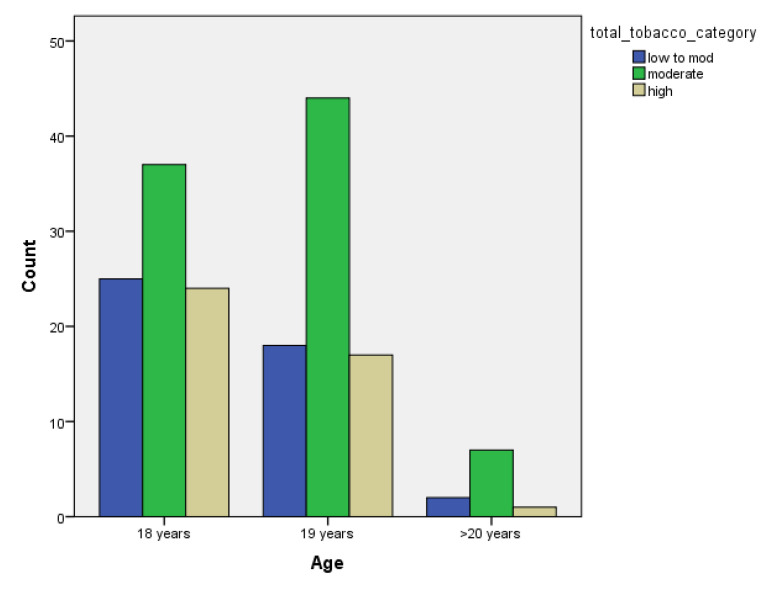
Frequency of tobacco and alcohol consumption depending on age (male).

**Figure 2 children-07-00071-f002:**
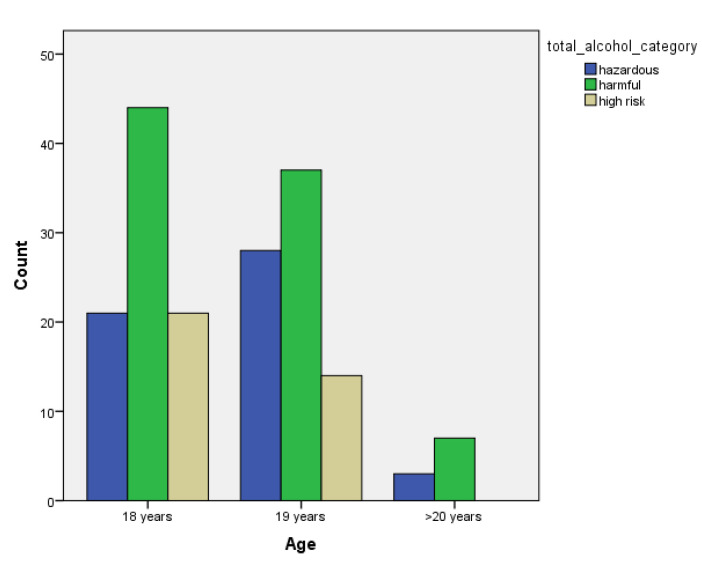
Frequency of tobacco and alcohol consumption depending on age (female).

**Figure 3 children-07-00071-f003:**
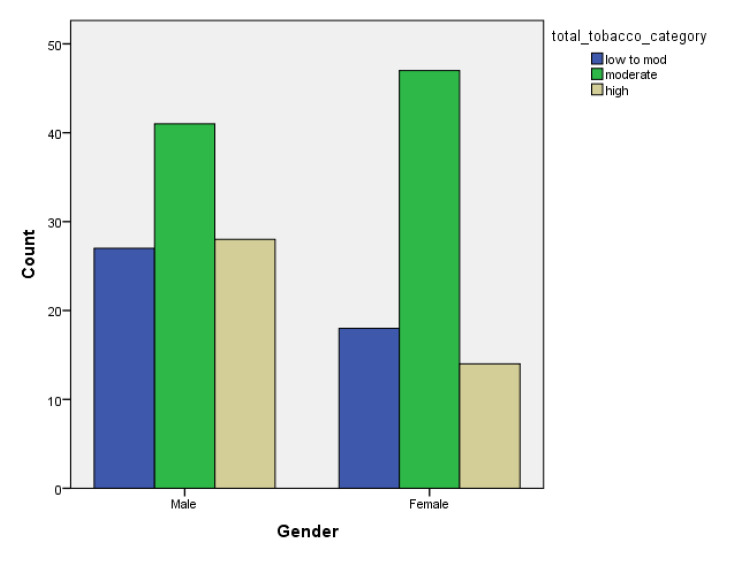
Frequency of tobacco and alcohol consumption depending on gender (male).

**Figure 4 children-07-00071-f004:**
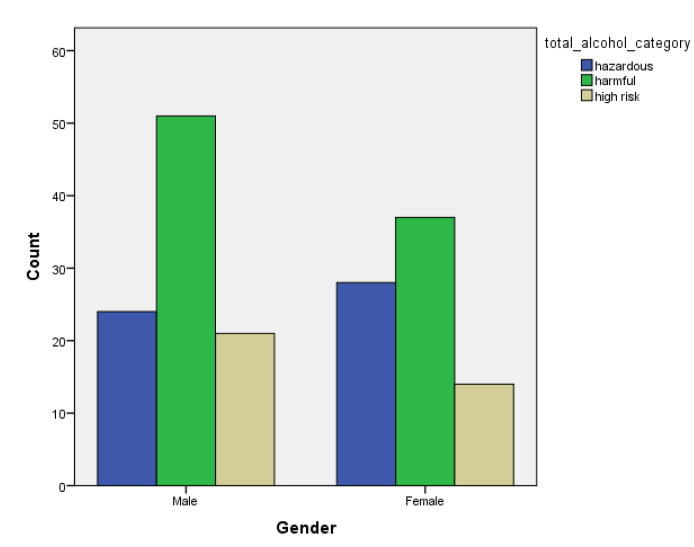
Frequency of tobacco and alcohol consumption depending on gender (female).

**Figure 5 children-07-00071-f005:**
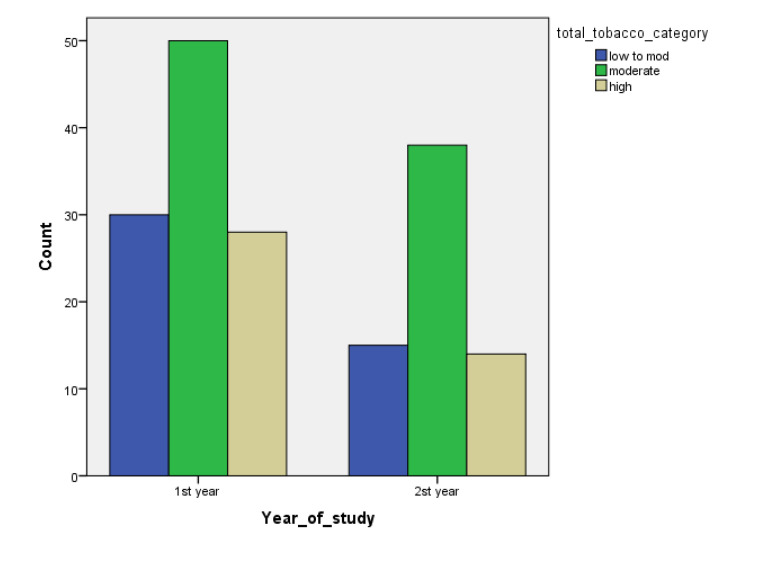
Frequency of tobacco and alcohol consumption depending on year of study (male).

**Figure 6 children-07-00071-f006:**
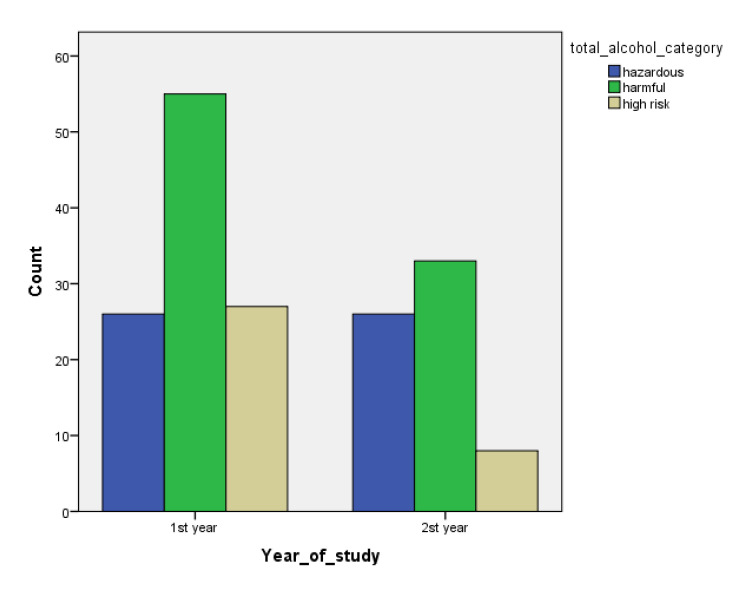
Frequency of tobacco and alcohol consumption depending on year of study (female).

**Figure 7 children-07-00071-f007:**
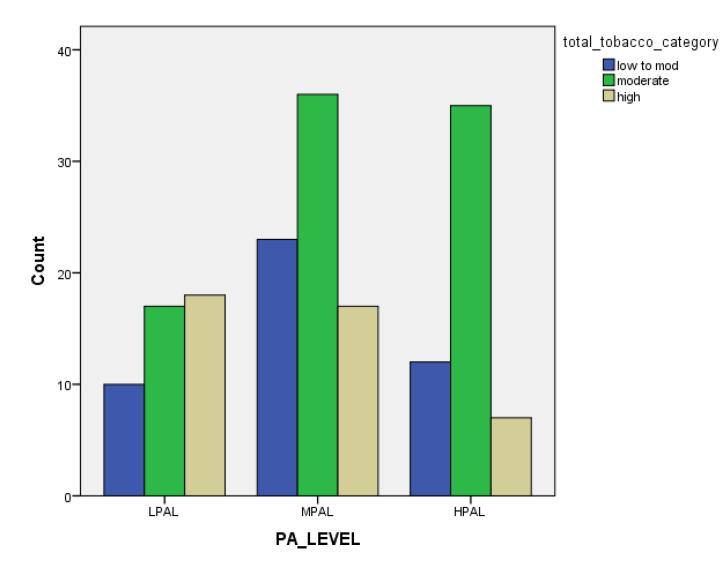
Frequency of tobacco and alcohol consumption depending on physical activity level (male).

**Figure 8 children-07-00071-f008:**
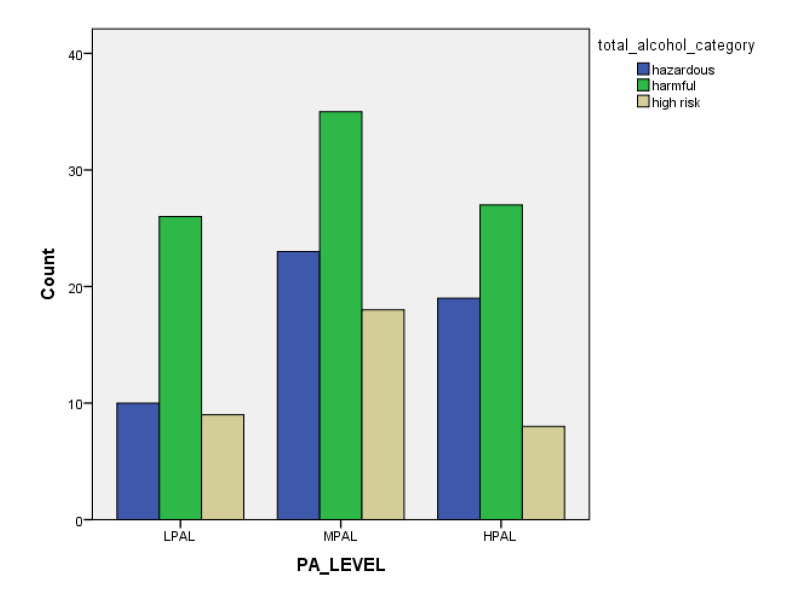
Frequency of tobacco and alcohol consumption depending on physical activity level (female).

**Figure 9 children-07-00071-f009:**
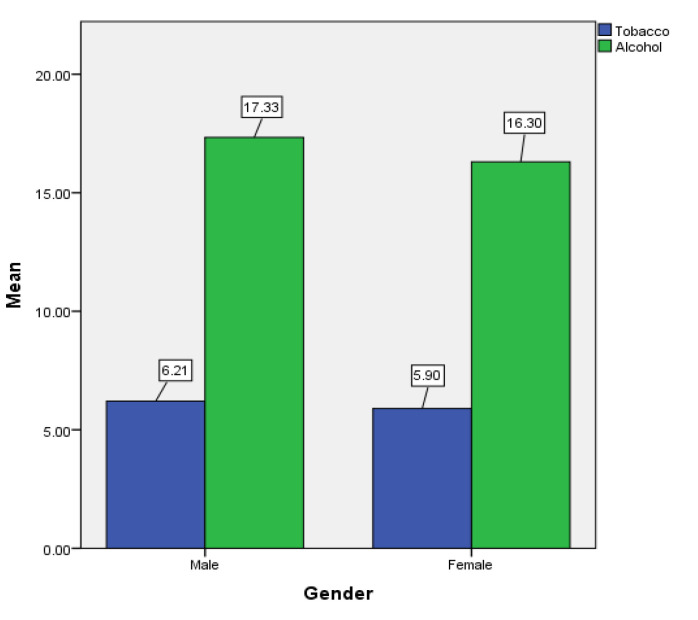
Gender differences for tobacco and alcohol consumption.

**Table 1 children-07-00071-t001:** Descriptive statistics of study variables.

Groups		Tobacco Consumption			Alcohol Consumption		
		Low	Moderate	High	Hazardous	Harmful	High Risk
*Gender*	*Male*	3.74 (0.44)	5.90 (0.53)	9.03 (0.79)	12.95 (2.49)	17.54 (1.18)	21.80 (1.93)
	*Female*	3.83 (0.38)	5.80 (0.64)	8.85 (0.77)	12.89 (2.31)	17.00 (1.00)	21.28 (1.43)
*Age*	*18 years*	3.72 (0.45)	5.89 (0.56)	9.00 (0.78)	12.85 (2.65)	17.65 (1.19)	21.80 (1.93)
	*19 years*	3.83 (0.38)	5.90 (0.60)	8.94 (0.82)	12.78 (2.25)	16.83 (0.89)	21.28 (1.43)
	*≥20 years*	4.00 (0.01)	5.28 (0.48)	9.00 (0.01)	14.66 (0.57)	17.71 (1.11)	-
*Year of study*	*1st year*	3.76 (0.43)	5.92 (0.56)	9.03 (0.79)	13.03 (2.42)	17.54 (1.18)	21.70 (1.83)
	*2ed year*	3.80 (0.41)	5.76 (0.63)	8.85 (0.77)	12.80 (2.36)	16.93 (0.96)	21.25 (1.48)
*PA*	*LPAL*	3.70 (0.48)	5.94 (0.55)	9.11 (0.75)	13.00 (2.78)	17.73 (1.15)	21.55 (1.66)
	*MPAL*	3.73 (0.44)	5.88 (0.57)	9.05 (0.82)	13.30 (1.98)	17.34 (1.16)	21.77 (1.95)
	*HPAL*	3.91 (0.28)	5.77 (0.64)	8.42 (0.53)	12.42 (2.61)	16.88 (0.97)	21.25 (1.48)

Abbreviations: PA—physical activity; LPAL—low physical activity level; MPAL—moderate physical activity level; HPAL—high physical activity level.

**Table 2 children-07-00071-t002:** Correlation coefficients of study variables.

Variable	Age	Year of Study	PA Level	Tobacco
Tobacco	−0.035	−0.032	−0.159 *	-
Alcohol	−0.193 *	−0.249 **	−0.179 *	−0.133

** *p* ≤ 0.01; * *p* ≤ 0.05.

**Table 3 children-07-00071-t003:** Tobacco and alcohol model fitting information.

Model	−2 Log Likelihood	Chi-Square	df	Sig.
Tobacco	141.400	14.963	6	0.021
Alcohol	49.225	13.895	6	0.031

Abbreviations: df – degree of freedom.

**Table 4 children-07-00071-t004:** Parameter estimates of tobacco and alcohol use.

		Estimate	Std. Error	Wald	df	*p*
Tobacco	[Age = 1.00]	−0.860	0.884	0.947	1	0.331
	[Age = 2.00]	0.454	0.624	0.531	1	0.466
	[Year of study = 1.00]	−1.370	0.716	3.662	1	0.056
	[Gender = 1.00]	1.097	0.763	2.069	1	0.150
	[PA_LEVEL = 1.00]	2.351	0.697	11.367	1	0.001 *
	[PA_LEVEL = 2.00]	1.202	0.565	4.532	1	0.033 *
Alcohol	[Age = 1.00]	1.023	0.951	1.156	1	0.282
	[Age = 2.00]	0.210	0.668	0.099	1	0.754
	[Year of study = 1.00]	2.499	0.796	9.855	1	0.002 *
	[Gender = 1.00]	−1.769	0.829	4.554	1	0.033 *
	[PA_LEVEL = 1.00]	−1.073	0.746	2.068	1	0.150
	[PA_LEVEL = 2.00]	−0.940	0.621	2.290	1	0.130

Abbreviations: PA—physical activity; * *p* ≤ 0.05. df – degree of freedom.

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
