# Peer review of "Predicting Tobacco and Alcohol Consumption Based on Physical Activity Level and Demographic Characteristics in Romanian Students"

_children, 2020, doi:10.3390/children7070071_

Round 1
Reviewer 1 Report
This paper meets the scientific conditions for its publication. The topic is of special relevance mainly because physical activity and sport is the best instrument for the prevention of the use and abuse of alcohol, tobacco and drugs at an early age.
Author Response
Dear Reviewer,
I attach the responses.
Thank you!

Reviewer 2 Report
The manuscript by Badicu et al. was motivated on the general harmful effects to health by behaviors such as smoking, alcohol over-consumption, and physical inactivity, particularly in young adults as they tend to persist for decades and thus lead to chronic diseases. The goal of the study was to explore if these three types of behavior were linked. The study was conducted recently (Nov-Dec of 2019) in a group of students of their home university and faculty, where about half of all the students (mostly first and second years) participated. All the measures of physical activity, smoking, and alcohol consumption were obtained by validated surveys, and analyzed by simple categorical associations. There are broad concerns about how this study was designed, data analyzed, and results interpreted, and conclusions generalized.
- According to Wikipedia, “BraÈ™ov has a population of 253,200 making it the 7th most populous city in Romania”. The University and its faculty of Physical Education and Mountain Sports is perhaps quite specialized in their country. Thus, how representative are these study participants as compare to other similar aged young adults in Transilvania, in this country, and a broader region? Given the statement in the Conclusions that all “… students requires that they always participate in PA”, do the findings of this study apply to anywhere else?
- To establish the potential of an age effect, the lack of inclusion of higher classes (3rd and 4th year students) was a major limitation. Moreover, the first two years of college are usually where big changes are occurring – students becoming independent, thus how much of the associations of some behaviors and the lack of the others were potentially impacted at this age?
- The study lacked basic subject characteristics (weight and height), measures socioeconomical status, and family history of smoking and alcohol, which could also influence the results.
- Physical activity (PA) was assessed for the past 7 days, but it was unclear if the alcohol and smoking surveys had the same restriction of time? PA was separated into three levels by the IPAQ; how were smoking and alcohol separated into three levels quantitatively?
- Did the authors demonstrate any associations between smoking and alcohol consumption (as the introduction seems to suggest)?
- What was the justification to use ordinal logistic regression vs. cluster analysis? Were there any clusters, especially between sexes? Table 1 should list men and women separately. Mysteriously, the total number of participants in this table was 175, not the 253 in the methods. What happened, especially there appeared to be an about 50% drop-out rate in female subjects?
- Why was p<0.001 chosen as the significance threshold in the data analysis section? If so, then the only statistically significant association in Table 3 was between “PA level 1” and “Tobacco”. Why were “Tobacco” and “Alcohol“ not controlled for each other in this table? Also, please keep the terminology consistent, e.g., smoking, cigarette smoking, tobacco use/consumption (they are not exactly the same).
- What are the units for the y-axes in Figures 1 and 2?
- Was PA significantly associate with smoking? The second sentence of the Discussion (line 162-3) seemed to directly conflict with the data and statement just presented above (line 157-9), Figure 2, and Table 3. Also, the concluding sentence in the Abstract also suggested significant associations between PA and smoking AND alcohol.
- Please discuss how regional/cultural differences may or may not explain the differences between this particular (homogeneous) population compare to others published in asking the similar questions in UK, Canada, Sweden, Spain, Brazil (Ref 37-41), and the US (Ref 46).
- There were statements about academic performances and cannabis in the introduction and discussions, which appeared to be outside of the goals of this study.
- What did the authors mean by “the types of tools” in line 214 of the Conclusions? How would different tools make a difference in the result?
- Please have a native or an experienced English speaker help with the writing.
Author Response

(The authors gave the same response as above.)

Reviewer 3 Report
This research is about the relationships between alcohol 73 consumption and smoking, PA, and demographic characteristics in students of the Faculty of 74 Physical Education and Mountain Sports in Brasov.
My primary concern about this study is authors did not explain the implication, limitation, and strength of the study. The discussion section needs to be improved.
Also, in the introduction section, authors need to explain about findings of other studies in relation to PA, alcohol, and smoking.
Why line no 52 is separated from the previous paragraph in the introduction?
In line 71, what specific factors authors are indicating?
In the methods section, can authors add a short description of the questions of the questionnaires? How did authors categorize tobacco (low, moderate, high) and alcohol use (hazardous, harmful, high risk)? What are the cut-off points?
Did the authors adjust for confounders in the analyses? If they did, a little description is needed.
Author Response

(The authors gave the same response as above.)

Reviewer 4 Report
Review of Manuscript children-785903
Title: Predicting of Tobacco and Alcohol Consumption by Physical Activity Level and Demographic Characteristics among Physical Education and Sport Sciences Students
Brief summary
The authors intend to show how physical activity (PA) and demographic characteristics relate to alcohol consumption and smoking. For this purpose they study a group of students admitted to the Faculty of Physical Education and Mountain Sports in Brasov, Romania. The results indicate somewhat higher tobacco and alcohol use among those with lower physical activity. Alcohol use but not tobacco use was higher among men than women.
Broad general comments
Strengths
- Choice of topic – The link between physical activity and other health behaviours is an interesting topic.
- Method – Good choice of measuring instruments. These have all been evaluated and applied in a number of earlier studies. Reliability was also tested specifically for each scale in relation to this sample.
- Analyses - Analysis methods are appropriate, although not argued for.
Weaknesses
- Introduction – could have discussed gender differences in alcohol and tobacco use as well as physical activity.
- Choice of population – Regarding the choice of study population, I am questioning how representative the students of physical education and sport science are of other young people or the general population? Especially since their subject of choice is the same as the interest of the study.
- Ethical issue – Although informed written consent was asked for in advance, I wonder how the researcher also being the students’ teacher (“applied by the researcher during his theoretical classes”) impact their responses to participate and how they respond to the research questions.
- Method - It is not explained how you reached your categories used in the result section. Cut-off points should be stated in the method section. Also, what were the incitements for the three categories? Rather, why does all alcohol refer to harmful use (hazardious, harmful, high risk) and tobacco to level of use (low, moderate, high)?
- Analyses – Socio-economic status could also have been a relevant demographic variable to look into in relation to these behaviours given that SES have been found to be related to differences in alcohol and tobacco use and also been observed to interact with gender.
- The results should be discussed more elaborately in the discussion section, e.g. regarding in terms of interpreting the results in relation to other countries where use/risk behaviours are likely to be less present (see more specific note).
- References – The literature on physical activity is fairly extensive in this paper, the number of references on alcohol and tobacco are much more limited. It would be useful to mention at least a few references from this research field, particularly regarding the relation between use of alcohol or tobacco and physical activity. For example paper by Kwan, Bobko, Faulkner, Donnelly & Caimey “Sport participation and alcohol and illicit drug use in adolescents and young adults: A systematic review of longitudinal studies” (2014, Addictive behaviours) and Lorente, Souville, Griffet & Grélot “Participation in sports and alcohol consumption among French adolescents” (2004, Addictive behaviours).
- References – Perhaps you could add references discussing differences between countries, i.e. comparative studies, in terms of alcohol and tobacco use and habits (e.g. results from the ECAS/European Comparative Alcohol Study) and differences by gender (e.g. results from GENACIS/Gender, Alcohol, and Culture: An International Study).
Specific comments
Introduction
- 2, row 55-56: “electronic devices that may be harmful for health”. In what way? Is there a reference for that statement? There is an ongoing international discussion of the decline in alcohol consumption among young people where increasing time on electronic devises and not the least in relation to gaming. Although I am not aware of a study which have been able to illustrate a strong effect on alcohol use, I don’t think your sentence can remain unchallenged.
p.2, row 62: the authors clearly argue for the importance of focusing on Romania, i.e. this country has a higher share of death attributable to these health risk behaviours than other EU states.
Materials and methods
- 3: What was the cut-off points for your indexes based on your scales? The total number should exceed three points.
- 3: What is the incitements behind the chosen categories? While using well-established measures, some of them could be categorized into more than three categories.
Results
- 4 & 5, row 143, 147 and 161: “coefficients of the model showed that only the PA level has the positive effect on tobacco use”, “On the other hand the alcohol consumption showed the different pattern” (results), “The key findings of this research showed different patterns of tobacco and alcohol consumption among physical education and sport sciences students.“ (discussion) – Couldn’t the diverged results for the two outcomes be related to how you measured the risk behaviours, i.e. your scales?
Discussion
- 5, row 166-171: “Our finding were consistent with El Ansari et al. [37], Dawson et al. [38], Von Bothmer et al. [39]. They had showed that females reported lower use alcohol, men engaged in more risky health behaviors than females (e.g. alcohol use) and female students had healthier habits related to alcohol consumption. On the other hand, our findings were inconsistent with Varela-Mato et al. [40], Pillon et al. [41] who reported that more female than male students use alcohol and tobacco. Also, Mays et al. [42] showed that alcohol use does not differ between men/boys and women/girls.”. It would be helpful to contextualize the results and other studies. What countries do the studies mentioned refer to? Are they different to Romania in terms of other aspects, e.g. alcohol/tobacco use levels and habits and gender differences?
- 6 , row 198-199: “Although the age periods, social and cultural differences of communities, the prevalence and prohibition of certain drugs in different countries and also different types of PA should be considered, which will be explained further below”. Following up on my previous comment above, I cannot see that you are discussing what you say you will discuss, why my comment remains.
Tables
Table 3 – the explanatory variables are not well illustrated, i.e. what is age=1 and age=2, et cetera. Tables should be able to stand alone.
Author Response
Dear Reviewer,
I attach the responses!
Thank you!

Round 2
Reviewer 2 Report
Thanks
Reviewer 4 Report
I can see that you have put in some work in response to my comments. I still have some concerns, though.
Comment 4 about gender differences – authors states that this have been addressed in the introduction, but nothing has been added, i.e. no references to earlier studies. The method and discussion section does, however, mention gender so it’s ok.
Comment 5 about representativeness of the sample – the authors write they agree with me but seems to have misunderstood my comment. Although about 80% of the students responded, you cannot claim that they are representative of the general population or even their age group. When one chose to study a particular subject, it goes without saying that one has a special interest in that subject. My point was therefore related to the fact that you have asked a group who clearly has an interest in physical activity questions about physical activity. Although you can claim that your sample maybe is representative of other young people with the same interest in sport/health, you cannot make the same assumption about other young people. This is easily addressed by you adding a sentence to the discussion that this group, due to their interest in physical activity, might differentiate from other groups of young people who do not share the same interest. Although not completely addressing my comment, a sentence on the representativeness have been added to the discussion.
Comment 6 about ethics – I still do not consider it completely voluntary if you are asked by your teacher to participate in a study who I understood is conducted by the teacher (?). The students are in a position of dependence and might agree to participate against their will. I let this one slide, it’s not a question for me as a reviewer but rather for the ethical board who agreed to the study in the first place.
Comment 8 about SES – socio-economic status refers to education, income and labour, not parents’ alcohol habits.
With that said, there have still been large improvements of the paper.